# Energy-Harvesting Powered Variable Storage Topology for Battery-Free Wireless Sensors †

**Firdaous El Mahboubi \*, Marise Bafleur, Vincent Boitier \* and Jean-Marie Dilhac**

LAAS-CNRS, Université de Toulouse, CNRS, INSA, UPS, 31042 Toulouse, France; marise.bafleur@laas.fr (M.B.); jean-marie.dilhac@laas.fr (J.-M.D.)

\* Correspondence: felmahbo@laas.fr or firdaous.el.mahboubi@laas.fr (F.E.M.); vboitier@laas.fr (V.B.)

† El Mahboubi, F.; Bafleur, M.; Boitier, V.; Dilhac, J.-M. Energy-Harvesting Powered Variable Storage Topology for Battery-Free Wireless Sensors. In Proceedings of the IEEE International Conference on Modern Circuits and Systems Technologies (MOCAST2018) on Electronics and Communications, Thessaloniki, Greece, 7–9 May 2018; doi:10.1109/MOCAST.2018.8376624.

**Abstract:** The energy autonomy of wireless sensors is one of the main roadblocks to their wide deployment. The purpose of this study is to propose simple adaptive storage architecture, which combined with energy harvesting, could replace a battery. The main concept is based on using several ultracapacitors (at least two) that are reconfigured in a series or in parallel according to its state of charge/discharge, either to speed up the startup of the powered system or to provide energy autonomy. The proposed structure is based on two ultra-capacitors, one of small capacitance value and one of big value. Powered by an energy-harvesting source, the devised control circuitry allows cold start up with empty ultra-capacitors, pre-regulated output voltage, and energy usage efficiency close to 94.7%.

**Keywords:** energy harvesting; autonomy; variable energy storage; ultra-capacitors; wireless sensor

---

## 1. Introduction

The energy autonomy of wireless sensors is one of the main roadblocks to their wide deployment in the different following areas: structural health monitoring (SHM) in severe environment such as aeronautics [1] or in building or infrastructure such as bridges with very long lifetime (>25 years) [2]. A way to strengthen the energy autonomy of these systems is to use energy harvesting from the surrounding environment coupled to a storage unit. The device used for storage is either a battery or an ultra-capacitor. In some cases, ultra-capacitors are more interesting than batteries: in conditions of extreme temperatures (in specific locations of an aircraft); in terms of safety (batteries may cause fire or explode); the peak power that can deliver ultra-capacitors is significantly larger than that of batteries (high power density); high charge/discharge efficiency and lifetime much greater (up to 500,000 cycles) than that of a battery (3000 to 4000 cycles). Even if the new technologies of low-power integrated circuits have made it possible to extend their lifetime, the replacement of hundreds or even thousands of batteries is not economically viable. Moreover, getting rid of primary batteries to avoid costly maintenance would be a must.

However, the storage in ultra-capacitor has some drawbacks and requires a compromise to satisfy two important objectives: a sufficient voltage during the initial charge must be rapidly reached (small capacitance) to get and maintain the powered system operational as quickly as possible and a large amount of energy should be stored (big capacitance) to increase its energy autonomy. A self-adaptive storage architecture consisting of four ultra-capacitors (UCs) was already proposed to address these constraints [3]. This structure is based on reconfiguring the storage elements from all

in-series to all in-parallel and reversely according to the availability of harvested energy and to the load consumption. This autonomous adaptive storage strategy allows a fast start-up and an increased energy usage (~10%) compared to a single big capacitor. However, it does not provide a pre-regulated voltage and induces abrupt voltage changes upon capacitor switching that could induce unwanted perturbations. We propose a variable storage topology constituted of only two ultra-capacitors, a small value one ($C_{small}$) and a large value one ($C_{big}$), which are appropriately switched to provide fast start-up of the system to be powered, large energy storage, output voltage pre-regulation, and autonomy of the system.

## 2. Basic Operation Principle

This adaptive storage system, aimed at supplying a wireless sensor node powered by ambient energy harvesting, is not new. Several solutions were already proposed in the literature: one needs a very large number of switches causing heavy losses [4], another requires a large number of ultra-capacitors [5], and a third one uses a complex architecture with multi-stacked dc-dc converters [6]. In this paper, we propose an alternative and very simple self-adaptive energy storage architecture. We analyze and experimentally compare two different architectures.

### 2.1. Basic Principle of Adaptive Storage

The proposed adaptive storage system, schematically described in Figure 1, is self-powered by an energy-harvesting source and is composed of at least two ultra-capacitors (UCs). It is meant to replace a battery. The basic idea is to adapt the size of the storage so that the system can start rapidly (small capacitance by implementing the different capacitor elements in series) but also provide a good autonomy (large capacitance by implementing the different capacitor elements in parallel). These configuration changes require a control circuitry that should be able to operate at low voltage, since it is supplied by the energy stored in the ultra-capacitors. The main challenges of this adaptive storage system are twofold:

- First, it should allow a cold startup operation that corresponds to the case when ultra-capacitors are empty.
- Second, its architecture should be optimized to minimize losses.

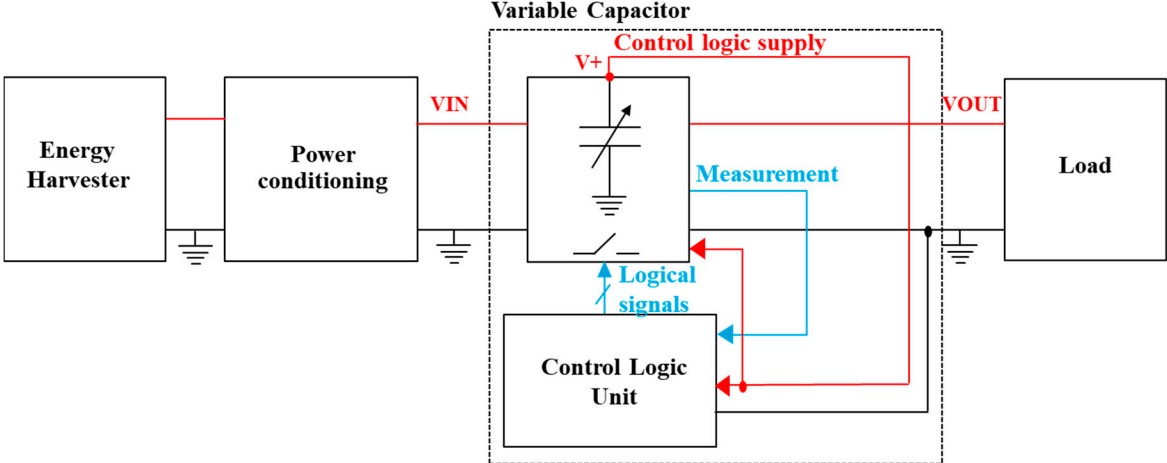

**Figure 1.** Block diagram of the proposed self-powered adaptive storage architecture (dashed-line block).

### 2.2. Reference Self-Adaptive Switched Architecture Storage with 4 Ultra-Capacitors

The reference architecture already proposed in a previous work [7] uses a matrix of four identical ultra-capacitors (Ci = C for I = 1, 2, 3, 4), interconnected by nine switches. As shown in Figure 2, three Schottky diodes allow a default serial structure at start-up.

During the charge phase, the change of configuration depends on the state of charge of the ultra-capacitor connected to ground ($V_{C4}$). This allows for three different configurations:

- "All in-series" allows fast startup ($C_{eq} = C/4$)
- "Series-parallel" ($C_{eq} = C$)
- "All in-parallel" in order to maximize the amount of stored energy ($C_{eq} = 4\,C$) without increasing V+ voltage.

And conversely for the discharge phase, to reduce the equivalent capacitance of the architecture and use the stored energy in ultra-capacitors as much as possible.

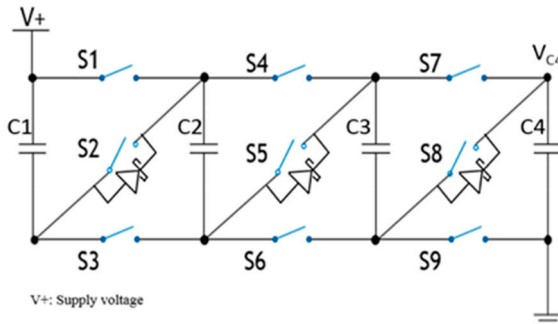

**Figure 2.** Electrical schematic of the reference self-adaptive switched architecture with 4 UCs (V+ = $V_{IN}$ = $V_{OUT}$).

For the sake of simplicity, this self-adaptive architecture storage does not include a balancing circuit, the maximum losses, related to the capacitance-value variability for a tolerance range of ±50%, being 2% of the relative stored energy. To characterize the efficiency of such reconfigurable storage, we have defined a figure of merit called the energy usage efficiency as follows:

$$\eta = \frac{E_{OUT}}{E_{IN}}, \tag{1}$$

where $E_{IN}$ is the total energy supplied to the adaptive storage unit and $E_{OUT}$ is the energy actually provided to the load.

The global losses including the required control electronics and the losses in the ultra-capacitors result in 93% energy usage efficiency. This autonomous structure insures a very fast start-up, stores a high amount of energy, and provides a maximum energy usage rate by limiting the residual energy left in the ultra-capacitor. However, it is quite complex and the abrupt voltage variation at each configuration change may induce electromagnetic disturbances. The purpose of the new proposed structure that is described in the following section is to reduce the complexity and at the same time, the power losses. In addition, this proposed structure exhibits less abrupt voltage variations and provides a pre-regulated voltage.

## 3. Proposed Self-Variable Storage Topology with 2 Ultra-Capacitors

The proposed storage topology requires only two ultra-capacitors, a small value one ($C_{small}$) and a large value one ($C_{big}$), which are appropriately switched to provide both a fast start-up of the system to be powered, large energy storage, and output voltage pre-regulation. The energy stored in a capacitor is $E_{stored} = \frac{1}{2}CV^2$. As a result, a given input power using a small value capacitor allows the requested voltage to reach across it much more rapidly than a single big capacitor.

### 3.1. Circuit Topology

The topology presented in Figure 3 is simple: it includes a Schottky diode to naturally ensure the initial charging of the small ultra-capacitor ($C_{small}$) and three switches, which allow the reconfiguration

of the two ultra-capacitors with regard to the supplied power and to the load. The Schottky diode provides a low threshold voltage and unidirectional flow for the current.

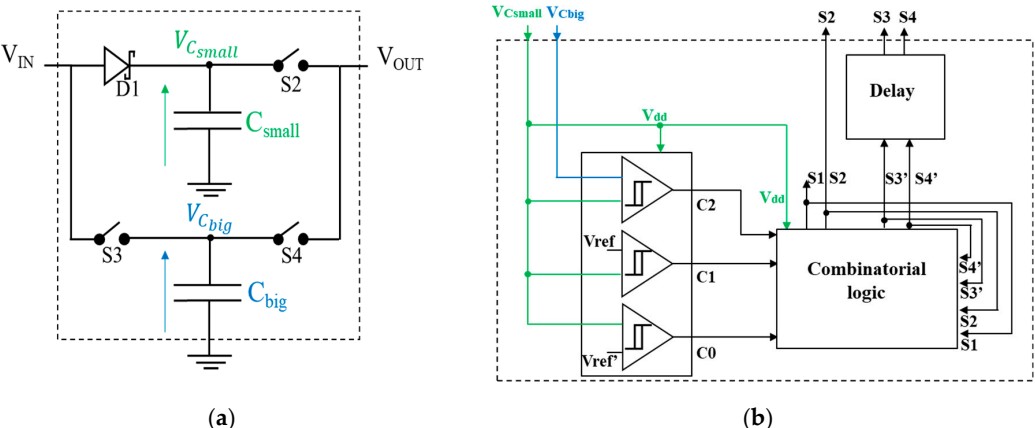

**Figure 3.** Electrical schematic of (**a**) the proposed variable storage architecture with 2 UCs and (**b**) its control circuitry.

### 3.2. Operating Principle

The principle of this variable storage architecture is to initially supply the small ultra-capacitor $C_{small}$ via D1 Schottky diode; all the switches being open (normally-off). The main objective is to be able to supply the load as soon as possible. A control circuitry monitoring the voltage across $C_{small}$, $V_{Csmall}$ and the voltage across $C_{big}$, $V_{Cbig}$ allows driving the switches according to appropriate voltage thresholds.

First, when $V_{Csmall}$ reaches a sufficient voltage (~0.8 V), the control circuitry is able to operate. In a second step, once $C_{small}$ has stored enough energy for supplying the load ($V_{max1}$ threshold), the control logic closes S2 switch to supply it (see Figures 4a,b and 5).

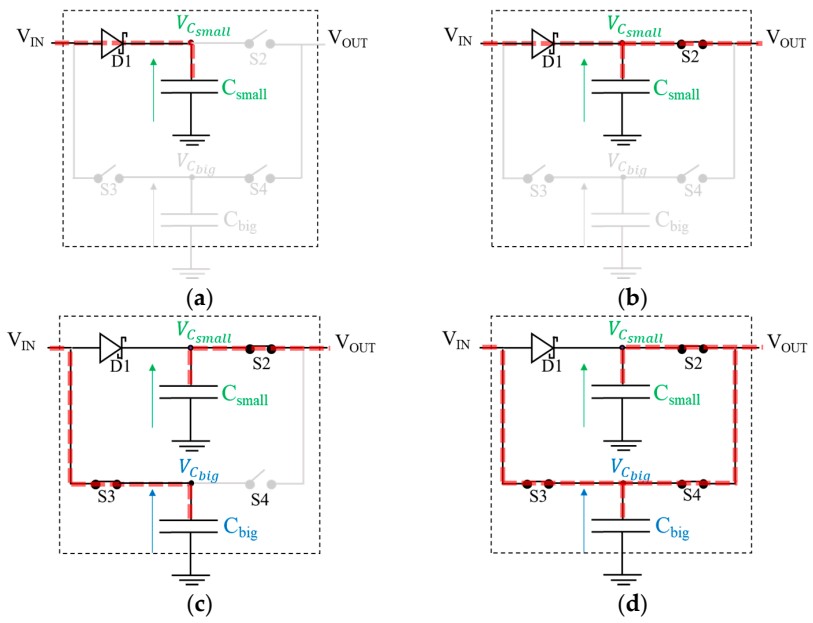

**Figure 4.** The different operation modes of the proposed variable storage architecture: (**a**) charge of $C_{small}$ only via D1 Schottky diode; (**b**) when reaching $V_{max1}$ threshold, S2 switch is closed and $C_{small}$ supplies the load; (**c**) $C_{small}$ continues to charge; when reaching $V_{max2}$ threshold, S3 switch is also closed to charge $C_{big}$ and $C_{small}$ supplies the load until reaching maximum discharge threshold $V_{min2}$; (**d**) when $V_{Csmall} = V_{Cbig}$, all switches are closed and both $C_{big}$ by $C_{small}$ supplies the load [8].

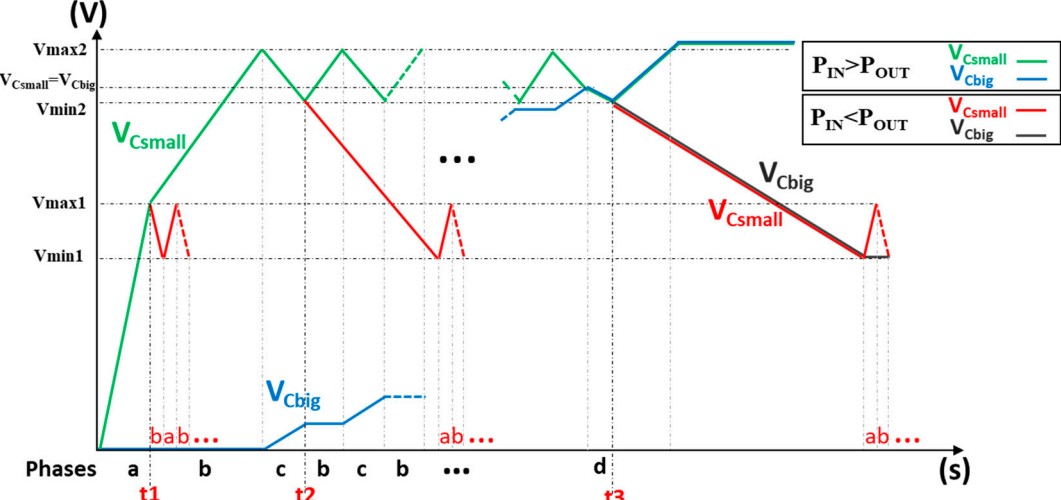

**Figure 5.** Operation principle of the proposed variable storage architecture for two scenario cases: harvested power ($P_{IN}$) $\geq$ consumed power ($P_{OUT}$) and harvested power ($P_{IN}$) < consumed power ($P_{OUT}$) (red curves, times t1, t2 and t3).

If the energy-harvesting source is sufficient, $V_{Csmall}$ can reach its maximum value ($V_{max2}$) and S3 switch is closed such that $C_{big}$ can be charged (see Figure 4c). This charge phase of $C_{big}$ is stopped as soon as $V_{Csmall}$ decreases by 50 mV ($V_{min2}$).

These two cycles (c and b) are repeated, the load being supplied by $C_{small}$, until the two voltages $V_{Csmall}$ and $V_{Cbig}$ become equal. In this case, the architecture moves to a parallel configuration (see Figure 4d) to maximize energy autonomy.

### 3.3. Logic Control Parameters and Optimization

For an optimized operation of this variable storage structure, two voltages, $V_{Csmall}$ and $V_{Cbig}$, need to be monitored and appropriate voltage thresholds defined. Moreover, two different cases of operations should be taken into account: first, when the harvested power ($P_{IN}$) is larger than the consumed power ($P_{OUT}$) and second, when the harvested power is smaller than the consumed power. Figure 5 illustrates these two cases. To control these switched capacitors, five different conditions are required:

$$V_{Csmall} \geq V_{max1}, \tag{2}$$

$$V_{Csmall} \leq V_{min1}, \tag{3}$$

$$V_{Csmall} \geq V_{max2}, \tag{4}$$

$$V_{Csmall} \leq V_{min2}, \tag{5}$$

$$V_{Csmall} = V_{Cbig}, \tag{6}$$

Let us start with the first ideal case being when harvested power ($P_{IN}$) is sufficient. As soon as $V_{Csmall}$ reaches $V_{max1}$ (the minimum voltage for which the load can be operated), S2 switch closes. While supplying the load, $V_{Csmall}$ continues to charge and reaches $V_{max2}$. Reaching this threshold means that the energy-harvesting source is sufficient and that $C_{big}$ can be charged. To do so, S3 switch is closed until $V_{Csmall}$ decreases to $V_{min2}$. The cycle is repeated ($V_{Csmall} \geq V_{max2}$ or $V_{Csmall} \leq V_{min2}$, see Figure 5), until $V_{Csmall} = V_{Cbig}$. At this phase, both ultra-capacitors operate in parallel to supply the load at a pre-regulated voltage around the average value of ($V_{max2} + V_{min2}$)/2. It is to be noted that in a real system to protect the ultra-capacitors, a voltage limitation (Zener diode or DC/DC converter) is required at the input of the adaptive storage unit.

If the energy-harvesting source becomes temporarily insufficient—see red curves in Figure 5 (at times t1, t2 and t3)—and the system is starting with empty ultra-capacitors, the startup is similar to the previous case and depending on the harvested power, only $C_{small}$ will be involved and as soon as it gets discharged down to $V_{min1}$, the S2 switch is opened so that it can get charged again up to $V_{max1}$. If the intermittency of the source occurs when $V_{Csmall} = V_{Cbig}$, both of them will continue supplying the load and get alternatively discharged down to $V_{min1}$ and charged up to $V_{max1}$.

To implement control circuitry, we used commercial discrete components: one Schottky diode, three bidirectional CMOS switches (normally-off ADG801), two LT6700 hysteresis comparators for ($V_{min1}$, $V_{max1}$) and ($V_{min2}$, $V_{max2}$), one LTC1540 comparator for the last condition (Equation (6)), eight AND gates, two inverters and two logical OR gates. To avoid detrimental short-circuit and related losses during configuration changes, we added a dead time (10 ms) to the logic signals applied to S1 and S3 switches. The sizing of the ultra-capacitors, $C_{small}$ = 100 mF and $C_{big}$ = 400 mF, was defined according to the chosen load, an interfacing LDO TPS78227 connected to a Jennic N5148-001 data logger.

To optimize the operating principle for this variable storage architecture, we performed electrical simulations using LTspice software. For the charging phase, we used a Thevenin generator made up of a 50 Ω resistor and a 5.1 V voltage source to simulate an energy-harvesting source. For the discharge phase, we connected a 1 kΩ resistor at the output.

During the different phases, we monitored the voltages at each intermediate node ($V_{Csmall}$, $V_{Cbig}$ and $V_{out}$, see Figures 6 and 7). The benefits of this variable architecture are fast start-up ($t_{ON}$~5 s) and high rate of stored energy with output pre-regulation.

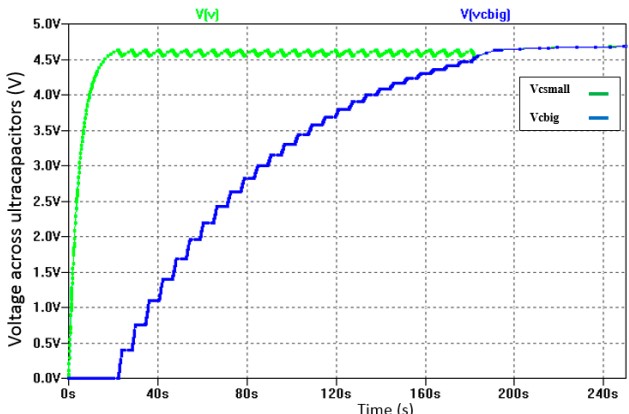

**Figure 6.** Simulated evolution of $V_{Csmall}$ and $V_{Cbig}$ voltages for the proposed variable storage architecture during charge and discharge for the ideal case (harvested power ≥ consumed power): energy source is a Thevenin generator ($E_{th}$ = 5.1 V, $R_{th}$ = 50 Ω) and a load resistor ($R_{LOAD}$ = 1 kΩ).

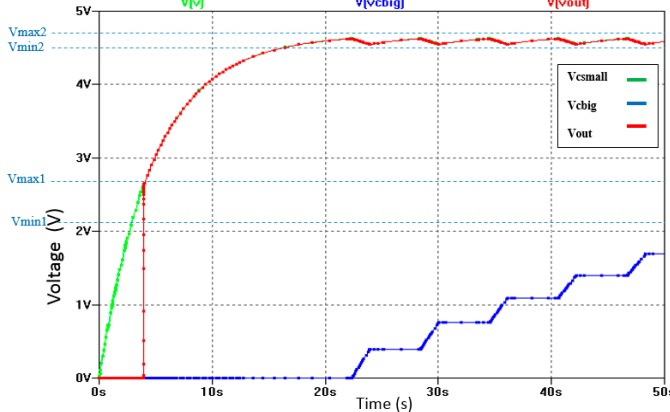

**Figure 7.** Zoomed view of Figure 6 showing details of the startup phase and the evolution of $V_{out}$.

### 3.4. Experimental Implementation and Characterization

A discrete prototype of the variable storage structure is shown in Figure 8. The electronic board on the left shows the adaptive storage architecture with $C_{small}$ and $C_{big}$ implemented with AVX BestCap ultra-capacitors. The electronic board on the right shows the control logic of this architecture.

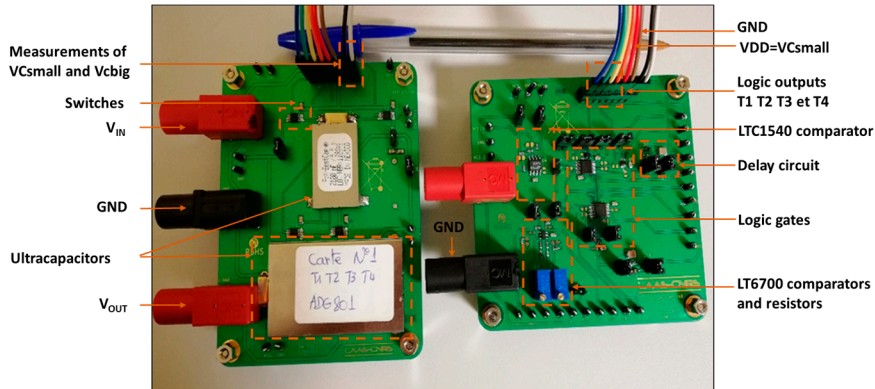

**Figure 8.** Prototype of the proposed variable storage architecture.

For the experimental validation, we defined the different voltage thresholds as follows: $V_{max1}$ = 2.8 V, $V_{max2}$= 4.4 V, $V_{min1}$ = 2.75 V and $V_{min2}$ = 4.35 V. Figure 9 provides the plots of the evolution of $V_{Csmall}$ and $V_{Cbig}$ for the ideal energy scenario when starting with empty ultra-capacitors. As soon as $C_{small}$ charging voltage reaches 0.8 V, the logic circuitry is activated and $V_{Csmall}$ and $V_{Cbig}$ are monitored. The global principle of the proposed variable storage architecture is validated. However, we detected an unexpected behavior at the very beginning: $C_{big}$ starts to get charged before $C_{small}$ reaches $V_{max2}$ thus delaying the powering of the load. We investigated this issue and found out two problems:

- The LTC1540 comparator exhibits an erroneous behavior when its VDD is below 1 V, thus inducing the triggering of S3 switch and the early charging of $C_{big}$. This behavior was not observed in simulation.
- The electrostatic discharge (ESD) protection of S3 switch provides a parasitic current path in parallel with the Schottky diode.

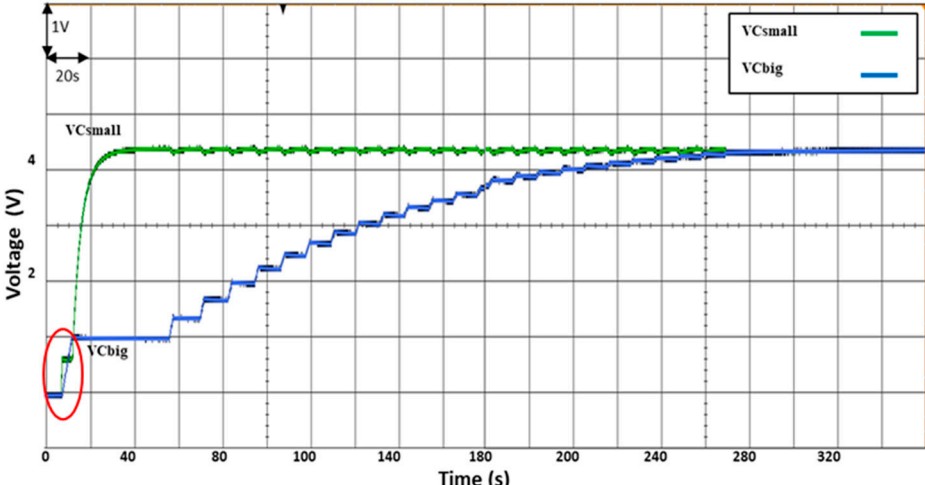

**Figure 9.** Measured behavior of the proposed variable storage architecture in the ideal case scenario: Energy source is a Thevenin generator ($E_{th}$ = 5.1 V, $R_{th}$ = 1 kΩ) and a load resistor ($R_{LOAD}$ = 1 kΩ).

The first issue was corrected by introducing a divider bridge by two before the input of S3 gate. It allowed getting the correct behavior of the variable storage structure as shown in Figure 10 in good agreement with the simulated one.

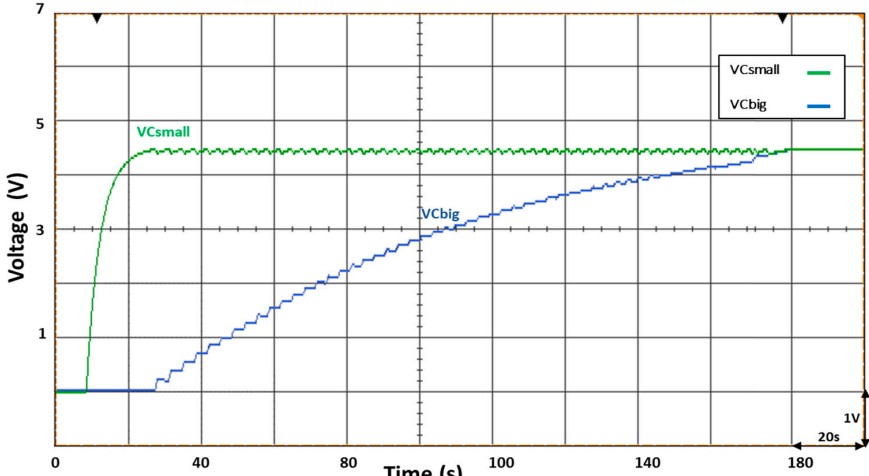

**Figure 10.** Measured behavior of the proposed variable storage architecture in the ideal case scenario after corrective action on the LTC1540 comparator. The energy source is a Thevenin generator ($E_{th}$ = 5.1 V, $R_{th}$ = 50 Ω) and a load resistor ($R_{LOAD}$ = 1 kΩ). The different voltage thresholds are as follows: $V_{max1}$ = 2.8 V, $V_{max2}$ = 4.45 V, $V_{min1}$ = 2.75 V and $V_{min2}$ = 4.4 V.

The second issue was identified when analyzing the respective losses of each block by LTspice simulation in the following conditions: charge phase, lasting 2 h 30 min, using a Thevenin source ($E_{th}$ = 5.1 V, $R_{th}$ = 50 Ω) as energy source and discharge phase, until reaching $V_{min1}$, with energy source disconnected and resistive load of 1 kΩ. They are summarized in Table 1. The energy usage efficiency $\eta$ as defined by Equation (1) is 91%. There are three main blocks that induce the majority of the losses: the switches, the Schottky diode and the ultra-capacitors. By analyzing the different current paths, it appeared that the high-power consumption of the switches is related to the parasitic path through the ESD protection diode of the S3 switch, which provides a less-resistive path compared to the Schottky diode whose on-resistance is 6 Ω. Implementing a Schottky diode with a much lower on-resistance such as the MBRS360 whose on-resistance is 0.042 Ω allows to greatly increase usage efficiency to up to 94.7%.

The losses in the ultra-capacitors are related to the parallel parasitic resistor associated with their self-discharge current. That means that this has to be a choice criterion for the ultra-capacitors implemented in such variable storage structure, in order to optimize performance.

**Table 1.** Energy consumption of each respective block in the variable storage structure computed by LTspice simulation for a total supplied energy of 210 J and energy provided to the load of 192 J.

| Block | Switches | Ultra-Capacitors | Schottky Diode | LT6700 Comparators | LT1540 Comparator | Logic Gates | Delay Circuits |
|---|---|---|---|---|---|---|---|
| **Losses** | 11 J | 3.9 J | 1.29 J | 330 mJ | 210 mJ | 3.6 mJ | 663 nJ |

*3.5. Final Optimization and Characterization*

Finally, to definitively get rid of any parasitic current path through the ESD protection of S3 switch and extend the voltage range of the adaptive storage unit, we decided to use discrete MOS switches and modify the architecture, as indicated in the electrical schematic of Figure 11. The main switch is a PMOS transistor that is driven by an NMOS transistor. A highly resistive resistor (7 MΩ) allows keeping it open as long as the NMOS transistor is inactive. The operation principle is almost the same as previously discussed and summarized hereafter:

a.    At start-up with empty ultra-capacitors, D1 charges $C_{small}$ and S2 and S3 are open.

b.    $V_{Csmall} \geq V_{max1} => $ S2 switch is closed and $C_{small}$ supplies the load.

c.    $V_{Csmall} \leq V_{min1} => $ S2 switch is opened, D1 recharges $C_{small}$.

d.    $V_{Csmall} \geq V_{max2} => $ S3 switch is closed and $C_{big}$ starts charging through D2 (Step c Figure 5).

e.    $V_{Csmall} \leq V_{min2} => $ S3 switch is opened, D1 recharges $C_{small}$. Steps b and c (Figure 5) are repeated until $V_{Cbig} = V_{Csmall}$.

f.    $V_{Cbig} \geq V_{Csmall} => $ D1, D2 and D3 diodes are ON, S2 and S3 switches are closed to supply the load from $C_{small}$ and $C_{big}$.

This proposed adaptive storage is easily scalable to higher power and voltage. Its advantage lies in the fact that the two ultra-capacitors are either perfectly isolated or when setup in parallel, charged at exactly the same voltage, avoiding any balancing current to flow. This is not the case for the 4 UCs-based structure where voltage monitoring is performed on one single UC. As a result, in the presence of capacitance variability (that could be as high as 20%), upon storage reconfigurability, balancing peak currents are going to flow. Anyway, for higher power, a soft start circuit for the switches would be needed to avoid any overstress of the UCs.

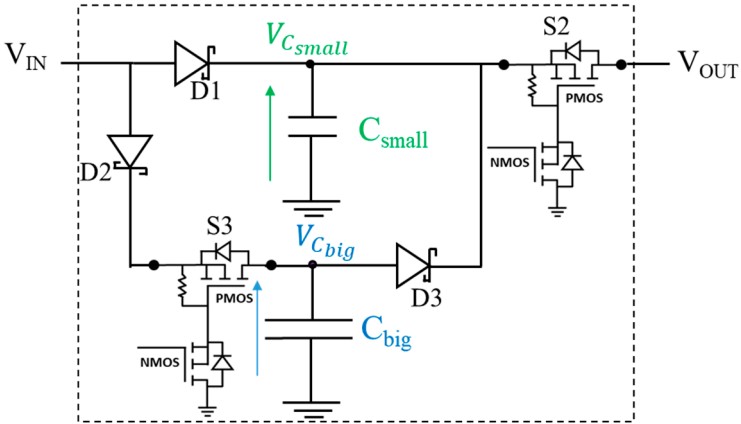

**Figure 11.** New version of the variable storage architecture using discrete MOS switches.

Figure 12 illustrates the measured behavior of the new variable storage architecture. It is very similar to the previous one, except that $C_{big}$ is able to charge to a higher voltage than $C_{small}$. This is due to the fact that $C_{small}$ is supplying the load whereas $C_{big}$ is only charging, due to the D3 diode that isolates it from the load as long as $V_{Cbig} < V_{Csmall} + V_{D3}$.

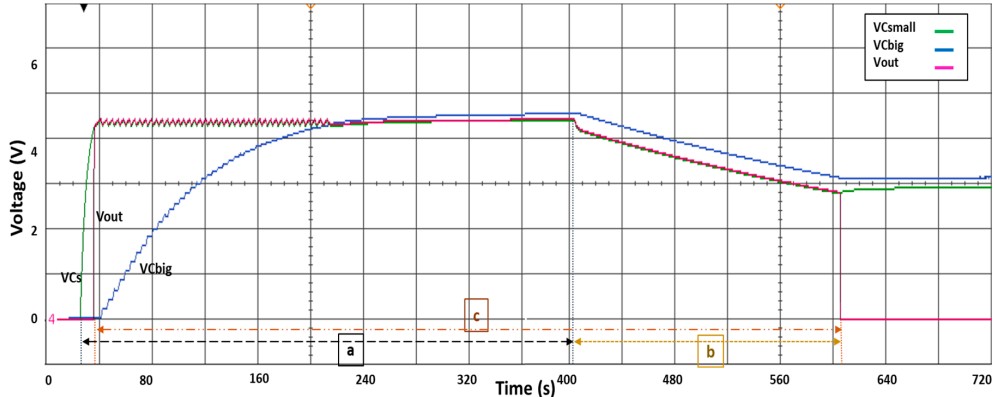

**Figure 12.** Experimental validation of the new version of the variable storage architecture using discrete MOS switches in the ideal case scenario: energy source is a Thevenin generator ($E_{th}$ = 5.1 V, $R_{th}$ = 50 Ω) and a load resistor ($R_{LOAD}$ = 1 kΩ). Defined voltage thresholds are: $V_{max1}$ = 4.2 V, $V_{max2}$ = 4.5 V, $V_{min1}$ = 2.8 V and $V_{min2}$ = 4.45 V. The Thevenin generator is disconnected at time t = 400 s (phase b).

We analyzed the resulting losses in this new architecture (Table 2) that are greatly improved compared to its previous versions, resulting in increased energy usage efficiency up to 94.6%. In addition, this new variable storage structure has the following advantages:

- Reduced number of components (control circuitry consists of only two comparators), resulting in lower cost and lower power consumption.
- Better genericness, meaning that it is able to operate at low and high voltages.
- Easily integrable in CMOS technology.

**Table 2.** Energy consumption of each respective block in the new variable storage structure computed by LTspice simulation for a total supplied energy of 216 J and energy provided to the load of 204.4 J.

| Block | Schottky Diodes | Ultra-Capacitors | LT1540 Comparators | Switches |
|---|---|---|---|---|
| **Losses** | 7 J | 4.08 J | 316 mJ | 310.9 mJ |

## 4. Implementation and Validation in a Wireless Sensor Node

To completely validate the concept of self-supplied adaptive storage, we implemented the second version of the proposed topology into a wireless sensor node, as shown in Figure 13. For a first test case, the energy-harvesting source is a photovoltaic cell ($V_{OC}$ = 1.5 V, $I_{SC}$ = 100 mA with a size of 50 mm × 68 mm) that is interfaced with a boost circuit BQ25504 to provide impedance matching, limiting the input voltage to 5.3 V. The load is a low-power Jennic 5148 datalogger that is interfaced with a TPS78227 Low Drop Out regulator (LDO) providing 2.7 V regulated voltage.

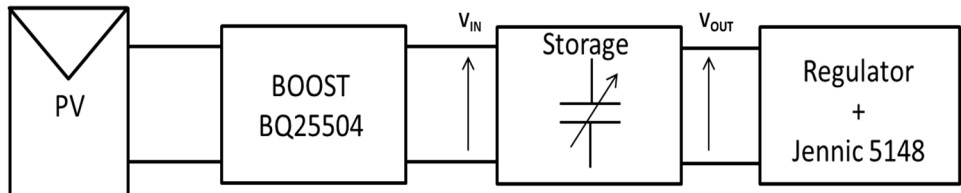

**Figure 13.** Implementation of the proposed variable storage architecture into a wireless sensor node powered by a photovoltaic cell (PV).

The datalogger performs temperature measurement every 47 s. Initialization startup energy of Jennic 5148 is ~0.5 J. The global energy consumption of one measurement cycle including communication and sleep mode is 13.8 mJ. Figure 14 provides the evolution of the input and output voltages of the adaptive UC when supplied by a photovoltaic cell over a full day. The startup time of the self-adaptive storage unit in irradiance around 720 W/m² is 30 s with ultra-capacitors completely empty. The autonomy during night is 5 h. It is interesting to note that the circuit starts up again in the morning for a very low irradiance around 25 W/m². In this full system, the global energy usage efficiency is dependent on the irradiance and is equal to 80% over the time interval (30 min); see Figure 14.

For a second test case, the energy-harvesting source is a much smaller photovoltaic panel ($V_{OC}$ = 5.6 V, $I_{SC}$ = 3 mA with a size of 50 mm × 20 mm). As its open circuit voltage $V_{OC}$ is compatible with UC handling voltage, we implemented a direct connection to the adaptive storage unit without any energy conditioning system. This test was performed for both the proposed adaptive storage ($C_{small}$ = 100 mF and $C_{big}$ = 2.5 mF) and a single ultra-capacitor ($C_{fixe}$ = 2.6 mF) of the same equivalent capacitance (corresponding to the equivalent capacitance of the adaptive storage unit at the end of charging phase). Compared to the previous case, we resized $C_{big}$ such that the supplied system could be fully autonomous under very low irradiance conditions. Each storage unit is connected to a photovoltaic panel under the same irradiance and supplies an LDO regulator TPS78227 and Jennic 5148 datalogger (as Figure 13). Figure 15 provides the evolution of the input and output voltages of

the adaptive UC in cold startup condition (UCs being empty). This adaptive storage makes it possible to validate a faster startup of 1 h 50 min compared to the single UC that needs 5 h 16 min, while ensuring the autonomy of the wireless system all night long. This startup may seem high but we have to take into account that at the beginning of this particular day, the solar panel provides a very low current and irradiation is also very low (cloudy day). The fluctuation of the input voltage (Figure 15) is linked to its connection to charge $C_{small}$ ultra-capacitor or to charge $C_{big}$ ultra-capacitor (permutation between Step b and c, Figure 4).

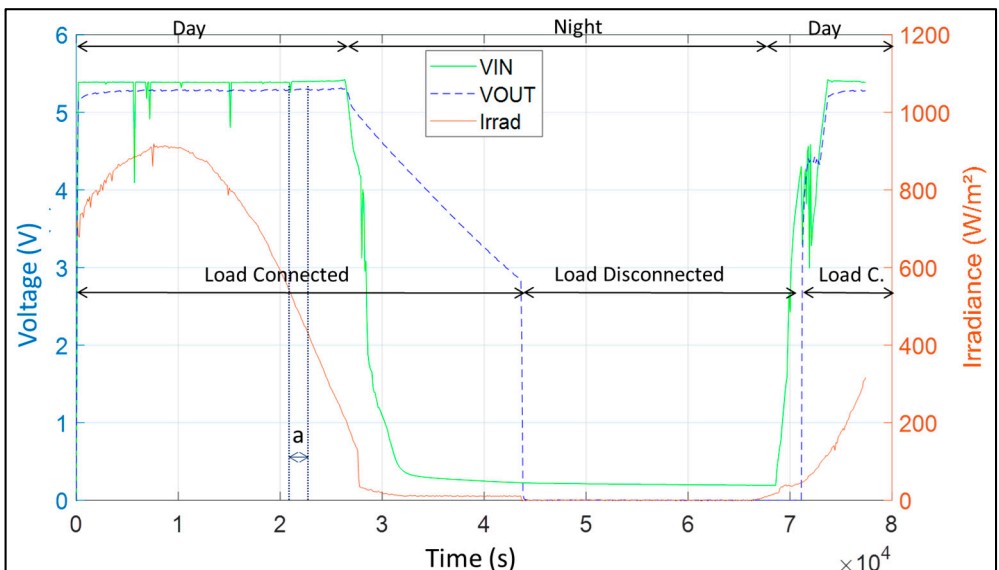

**Figure 14.** Implementation of the proposed variable storage architecture (with $C_{small}$ = 100 mF and $C_{big}$ = 400 mF) into a wireless sensor node powered by a photovoltaic cell (PV): evolution of irradiance, input and output voltages of adaptive UC over one day.

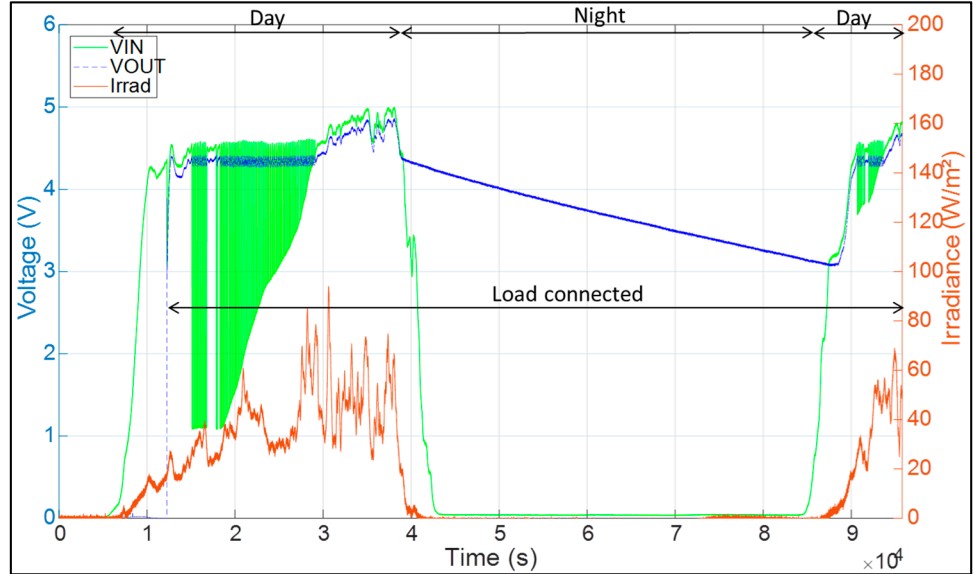

**Figure 15.** Implementation of the proposed variable storage architecture (with $C_{small}$ = 100 mF and $C_{big}$ = 2.5 F) into a wireless sensor node powered by a photovoltaic cell (PV): evolution of irradiance, input and output voltages of adaptive UC over one day and a half with very bad meteorological conditions.

## 5. Discussion

In Table 3, the proposed adaptive storage architecture is compared to the reference architecture with four UCs and to one from the literature [9]. We can see that both versions of the proposed structure allow increasing energy usage efficiency, thanks to simplification of the circuitry and the reduction of the number of UCs and switches. Compared to the other structures proposed in the literature, it is self-supplied from an energy-harvesting source, can operate in cold startup i.e., when the UCs are empty, and provides at its output a pre-regulated voltage. Compared to the four UCs-based structure of Figure 2, the output voltage undergoes variations at a very slow rate, thus resulting in electromagnetic emissions. The residual voltage is very dependent on the voltage regulator setup at the output; some of them are operational with very low input voltage (<<1 V). In addition, the sizing of the big capacitor, defining the energy autonomy of the powered system, can be independent of the sizing of the small one that defines the startup speed. It is well suited for CMOS integration, and when combined with silicon-integrated microsupercapacitors [10] makes a very attractive and compact solution to the replacement of a battery.

**Table 3.** Comparison of the proposed adaptive storage to previously published ones.

| Topology/Performance | Reference Topology with 4 UCs (Figure 2) | 1st Version of Proposed Topology with 2 UCs (Figure 3) | 2nd Version of Proposed Topology with 2 UCs (Figure 11) | Topology with 4 UCs from [9] |
|---|---|---|---|---|
| Complexity: UCs/Switches/Schottky diode/Comparators/Logic gates | 4/9/3/2/2 | 2/3/1/3/12 | 2/2/3/2/0 | 4/5/Not available |
| Low variations of output voltage | - | +++ | +++ | - |
| Fast startup | +++ | +++ | +++ | +++ |
| Pre-regulated output voltage | - | ++ | ++ | - |
| Low cost | + | ++ | ++ | + |
| Residual energy | + | - | - | + |
| Self-supply | + | + | + | - |
| Autonomy (Ratio Cmax/Cmin) | 16 | As desired | As desired | 4 |
| Energy usage efficiency | 93% | 94.7% | 94.6% | 88% |

**Author Contributions:** Conceptualization, M.B., V.B., J.-M.D. and F.E.M.; methodology, F.E.M.; validation, F.E.M.; formal analysis, F.E.M.; investigation, F.E.M.; supervision, M.B. and V.B.; project administration, J.-M.D.; funding acquisition, M.B. and J.-M.D.

**Funding:** This research work was partially carried out within the framework of the European project SMARTER funded by the CHIST-ERA program, "Green ICT, towards Zero Power ICT".

**Acknowledgments:** This work is carried out within the framework of the European project SMARTER funded by the CHIST-ERA program, "Green ICT, towards Zero Power ICT".

**Conflicts of Interest:** The authors declare no conflict of interest.

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
