# Peer review of "Energy-Harvesting Powered Variable Storage Topology for Battery-Free Wireless Sensors†"

_technologies, doi:10.3390/technologies6040106_

Reviewer 1 Report

The paper "Energy-Harvesting Powered Variable Storage Topology For Battery-Free Wireless Sensors" describes an active system built from two different UCs whose task is to quickly achieve and maintain a certain voltage level to power the portable sensors.

The authors clearly describe the successive stages of the formation of system’s concept, assumptions, appropriate topology, necessary improvements, and test results of the prototype system. In principle, the article could be published in its current form. However, due to the fact that the journal "Technologies" is addressed to readers from various fields of science and technology, in order to increase interest in the article in a wider range of readers, and then to increase its citation, I propose the following improvements:

1)      I suggest a wider discussion (additional 2-3 sentences) why the authors give up the battery. There are known applications, where the battery provides energy for many years (eg pacemakers).

2)      I suggest giving more examples of using the proposed system in real applications, to give the readers an idea, inspiration about the possibilities of using this system (or a similar one) for own applications.

3)      I suggest a wider discussion (additional 2-3 sentences) about the scalability of the system, whether it is limited only to single Watt power, or could it be used on larger scales (powers of kW) - in which applications and under what conditions.

4)      Are there problems due to the difference in UC voltages and high currents during switching? - In larger circuits before connecting the UC or to batteries, voltage equalization or kind of "soft start" is required. Does it not affect the durability of the switches?

5)      Did the authors study / consider the behavior of the system under different environmental conditions (temperature, humidity, vibrations)? Would it be necessary? Would this affect the efficiency?

6)      The authors use the abbreviation "mn" for the word "minute". In my opinion, "min" should be used. Authors use in line 266 the units "cm". It will be more technically correct to use "mm"

7)      Figure 15 presents a non-standard case, therefore its description in text should be more detailed, to help the reader interpret the data presented in it.

8)      References should contain more items and should be presented in a unified form.

Author Response

Point 1: I suggest a wider discussion (additional 2-3 sentences) why the authors give up the battery. There are known applications, where the battery provides energy for many years (eg pacemakers).

Response 1:

The device used for storage, is either a battery or an ultra-capacitor. In some cases, ultra-capacitors are more interesting than batteries: in conditions of extreme temperatures (in specific locations of an aircraft); in terms of safety (batteries may cause fire or explode); the peak power that can deliver ultra-capacitors is significantly larger than that of batteries (high power density); high charge/discharge efficiency and lifetime much greater (up to 500 000 cycles) than that of a battery (3000 to 4000 cycles). Even if the new technologies of low-power integrated circuits have made it possible to extend their lifetime, the replacement of hundreds or even thousands of batteries is not economically viable. Moreover, getting rid of primary batteries to avoid costly maintenance would be a must.

Point 2: I suggest giving more examples of using the proposed system in real applications, to give the readers an idea, inspiration about the possibilities of using this system (or a similar one) for own applications.

 Response 2:

We added this sentence and bibliography:

The energy autonomy of wireless sensors is one of the main roadblocks to their wide deployment, in the different following areas: structural health monitoring (SHM) in severe environment such as aeronautics [1] or in building or infrastructure such as bridges with very long lifetime (>25 years) [2].

1.       Bafleur M.; Boitier V.; Bramban D.; Dilhac J-M.; Dollat X.; Féau J.; Jugé S. Autonomous power supply for aeronautical health monitoring sensors. J. Phys. Conf. Ser., vol. 1052, no 1, p. 012031, 2018.

2.       Kim S.; Pakzad S.; Culler D.; Demmel J. ; Fenves G.; Glaser S.; Turon M. Health Monitoring of Civil Infrastructures Using Wireless Sensor Networks. In 2007 6th International Symposium on Information Processing in Sensor Networks, 2007, p. 254‑263.

Point 3: I suggest a wider discussion (additional 2-3 sentences) about the scalability of the system, whether it is limited only to single Watt power, or could it be used on larger scales (powers of kW) - in which applications and under what conditions.

 Response 3:

This proposed adaptive storage is easily scalable to higher power and voltage. Its advantage lies in the fact that the two ultra-capacitors are either perfectly isolated or when setup in parallel, charged at exactly the same voltage, then avoiding any balancing current to flow. This is not the case for the 4 UCs-based structure where voltage monitoring is performed on one single UC. As a result, in presence of capacitance variability (that could be as high as 20%), upon storage reconfigurability, balancing peak currents are going to flow. Anyway, for higher power, a soft start circuit for the switches would be needed to avoid any overstress of the UCs.

Point 4: Are there problems due to the difference in UC voltages and high currents during switching? - In larger circuits before connecting the UC or to batteries, voltage equalization or kind of "soft start" is required. Does it not affect the durability of the switches?

 Response 4: See above

Point 5: Did the authors study / consider the behavior of the system under different environmental conditions (temperature, humidity, vibrations)? Would it be necessary? Would this affect the efficiency?

 Response 5: The purpose of this work was to make a proof of concept. Of course, temperature will slightly affect the efficiency.

Point 6: The authors use the abbreviation "mn" for the word "minute". In my opinion, "min" should be used. Authors use in line 266 the units "cm". It will be more technically correct to use "mm"

 Response 6: We corrected this in the article.

Point 7: Figure 15 presents a non-standard case, therefore its description in text should be more detailed, to help the reader interpret the data presented in it.

 Response 7:  

We added the following description:

For a second test case, the energy-harvesting source is a much smaller photovoltaic panel (VOC= 5.6V, ISC=3 mA with a size of 50 mm*20 mm). As its open circuit voltage, VOC is compatible with UC handling voltage, we implemented a direct connection to the adaptive storage unit without any energy conditioning system. This test was performed for both the proposed adaptive storage (Csmall=100 mF and Cbig=2.5 mF) and a single ultra-capacitor (Cfixe=2.6 mF) of the same equivalent capacitance (corresponding to the equivalent capacitance of the adaptive storage unit at the end of charging phase). Compared to the previous case, we have resized Cbig so as the supplied system could be fully autonomous under very low irradiance conditions. Each storage unit is connected to a photovoltaic panel under the same irradiance and supplies an LDO regulator TPS78227 and Jennic 5148 datalogger (as Fig. 13). Figure 15 provides the evolution of the input and output voltages of the adaptive UC in cold startup condition (UCs being empty). This adaptive storage makes possible to validate a faster startup of 1h50min compared to the single UC that needs 5h16min whereas ensuring the autonomy of the wireless system all night long. This startup may seem high but we have to take into account that at the beginning of this particular day, the solar panel provides a very low current and the irradiation is also very low (cloudy day). The fluctuation of the input voltage (Figure 15) is linked to its connection to charge Csmall ultra-capacitor or to charge Cbig ultra-capacitor (permutation between step b and c, Figure 4).

Point 8: References should contain more items and should be presented in a unified form.

 Response 8: We corrected this in the article.

Reviewer 2 Report

Topologies proposed in [2], [5], [6] have been proposed to charge/discharge capacitors deeply while maintaining their output voltage in a certain narrow range. The output voltage of the authors' proposed circuit varies significantly during discharging. Isn't this a major drawback of the proposed circuit? Please discuss about it.

Author Response

Point 1: Topologies proposed in [2], [5], [6] have been proposed to charge/discharge capacitors deeply while maintaining their output voltage in a certain narrow range. The output voltage of the authors' proposed circuit varies significantly during discharging. Isn't this a major drawback of the proposed circuit? Please discuss about it.

 Response 1:

Compared to the 4 UCs-based structure, advantages of the proposed 2 UCs-based structure:

·       a slow variation rate : no EMI

·       residual voltage is very dependent on the voltage regulator setup at the output. Some of them are operational with very low input voltage (<<1V)
